# Cost-effectiveness of physical activity interventions in adolescents: model development and illustration using two exemplar interventions

Vijay S Gc, [1,2] Marc Suhrcke, [1,3] Andrew J Atkin, [4] Esther van Sluijs, [5] David Turner [2]

¹Centre for Health Economics, University of York, York, UK
²Norwich Medical School, University of East Anglia, Norwich, UK
³Luxembourg Institute of Socio-Economic Research, Esch-sur-Alzette, Luxembourg
⁴Faculty of Medicine and Health Sciences, University of East Anglia, Norwich, UK
⁵MRC Epidemiology Unit, University of Cambridge, Cambridge, UK

**Correspondence to**
Vijay S Gc; vijay.gc@york.ac.uk

## ABSTRACT

**Objective** To develop a model to assess the long-term costs and health outcomes of physical activity interventions targeting adolescents.

**Design** A Markov cohort simulation model was constructed with the intention of being capable of estimating long-term costs and health impacts of changes in activity levels during adolescence. The model parameters were informed by published literature and the analysis took a National Health Service perspective over a lifetime horizon. Univariate and probabilistic sensitivity analyses were undertaken.

**Setting** School and community.

**Participants** A hypothetical cohort of adolescents aged 16 years at baseline.

**Interventions** Two exemplar school-based: a comparatively simple, after-school intervention and a more complex multicomponent intervention compared with usual care.

**Primary and secondary outcome measures** Incremental cost-effectiveness ratio as measured by cost per quality-adjusted life year gained.

**Results** The model gave plausible estimates of the long-term effect of changes in physical activity. The use of two exemplar interventions suggests that the model could potentially be used to evaluate a number of different physical activity interventions in adolescents. The key model driver was the degree to which intervention effects were maintained over time.

**Conclusions** The model developed here has the potential to assess long-term value for money of physical activity interventions in adolescents. The two applications of the model indicate that complex interventions may not necessarily be the ones considered the most cost-effective when longer-term costs and consequences are taken into account.

## INTRODUCTION

Insufficient physical activity is a key risk factor for chronic diseases, such as cardiovascular disease, type 2 diabetes, and some types of cancer in the general population.[1 2] Physical activity in young people is associated with many health benefits including improved cardiovascular and mental health,[3 4] academic performance[5] and bone health.[6] While

### Strengths and limitations of this study

► A Markov cohort model was developed based on currently available evidence to simulate the long-term impacts in terms of costs and quality-adjusted life years of physical activity interventions for adolescents.
► The study incorporates the most recent evidence on the effect of increased physical activity in long-term chronic disease conditions.
► The model builds on previously published cohort models and includes additional health states. In addition, extensive sensitivity analyses have been performed to reflect uncertainty in model structure and parameter assumptions.
► A limitation of the present study is that the change in activity level over time were estimated using population-level prevalence data due to unavailability of longitudinal data describing the lifetime trajectory of physical activity and exclusion of long-term impacts on other conditions, for example, mental health.

physical activity typically declines with age, active children are more likely to become active adults.[7 8] Although the short-term and long-term health benefits of physical activity are well-documented,[9 10] in England, nearly half of all young people fail to achieve the recommended levels of physical activity, based on self-reports.[11 12] When measured objectively using accelerometers, the prevalence of inactivity is higher still (91% boys and 98% girls).[13]

The high prevalence of physical inactivity in young people places a significant burden on healthcare services and the wider economy. A 2014 report estimated a lifetime cost of £53.3 billion related to inactivity among today's 11–25 years old,[12] taking into consideration the fact that physical activity levels in childhood predict adult activity levels.[14] This estimate includes direct healthcare costs of treating the burden of type 2 diabetes, coronary heart disease (CHD), stroke and colon

cancer, and the risk of premature death and morbidity associated with these illnesses.

In recent years, there has been increasing interest in identifying interventions to improve young people's activity levels. Although some school-based physical activity interventions show promising effects[15][16] the existing evidence is very limited in both quantity and quality.[17] While improvements in physical activity may have long-term health benefits, the evidence on the longer-term costs and health benefits of interventions in adolescence is particularly sparse. Trials generally do not have sufficient follow-up to capture associated longer-term costs and consequences directly.[18] Quantifying the economic and health benefits associated with physical activity interventions would help decision makers to make informed decisions, that is, assessing whether these interventions are an efficient use of limited healthcare resources.

Furthermore, much of the health benefits of physical activity interventions occur in the future. Also, many interventions are focused on adult or elderly populations. The long-term costs and health benefits of physical activity interventions in an adolescent population are a comparatively scarcely researched area. To fill this critical research gap, we developed a decision analytic model aimed at quantifying the potential long-term costs and quality-adjusted life-year (QALY) implications of changes in activity levels during adolescence. We then illustrate some of the practical implications of taking a longer-term perspective by applying the model to two exemplar intervention programmes to show how the changes in levels of adolescent physical activity could affect activity levels throughout lifetime, as well the resulting longer-term costs and health benefits.

## METHODS
We developed a probabilistic, age-dependant and gender-dependant state-transition Markov model to simulate a cohort of healthy adolescents. The model estimates the risk of cardiovascular, type 2 diabetes and oncological events over a lifetime, and associated costs and quality of life. The model structure was based on the previously published models[19][20] that assessed the cost-effectiveness of physical activity interventions in the adult population. The model combines information from a variety of sources relating to disease and physical activity epidemiology, mortality, effectiveness, health-related quality of life and costs.

### Structuring the model
#### Model and population type
A simulated cohort of 10 000 healthy adolescents aged 16 entered the model. The intervention is assumed to have been delivered at the start of the first model cycle. At the end of the first cycle, based on the intervention effectiveness evidence, a proportion of cohort members move to a higher activity level. Depending on the sustainability of the intervention effect, in subsequent years cohort members obtain an annual probability of remaining at the new activity level or moving to a lower physical activity state or to a disease state or death.

### Model states
The health states included in the model are 'healthy' (disease-free), 'having a chronic disease' and 'dead'. At the beginning of the simulation, we assumed that all cohort members start out as healthy, that is, disease free. Within the model, physical activity is classified into four activity levels (inactive, low, moderate and high) based on weekly moderate-to-vigorous physical activity (MVPA). The model has 11 health states in total: four physical activity levels, six chronic disease conditions, and a dead state (figure 1). Among the healthy, the risk of developing one of the six diseases depends on age, gender and activity level. For simplicity, we assumed that health states included in the model were mutually exclusive, and cohort members did not move between disease states.

The selection of the disease conditions was based on currently available evidence describing the association between physical activity and disease risk,[21–26] and economic and health burden of diseases related to physical inactivity in the UK.[27] Disease conditions are all associated with costs and impact on quality of life. For each of the selected exemplar interventions, we re-ran the model changing appropriate data to reflect the costs and effectiveness of these interventions. Costs and QALYs were discounted at 3.5% as recommended for the National Institute for Health and Care Excellence (NICE) reference case.[28]

### Time horizon
The time horizon of the model is 65 years, that is, the model follows the cohort of 16 years old until they reach 81 years—the average life expectancy in the UK.[29] A half-cycle correction was applied under the assumption that each transition happened halfway during the cycle.[30]

### Populating the model
#### Baseline population and activity levels
Data on age and gender distribution of the initial population were obtained from the Office for National Statistics (ONS).[31] An estimate of baseline activity level (weekly MVPA) was taken from the 2012 Health Survey for England (HSE).[32] Participants were divided into four levels (<30, 30–149, 150–420 and ≥421 MVPA min per week) of activity by age and gender, based on the UK Department of Health's physical activity recommendations.[11] The moderate activity category equates to the current recommended level of physical activity.

#### Transition probabilities
To estimate the annual probability of developing each disease, the annual incidence rates for the disease conditions included in the model were taken from population-based studies in the UK.[33–39] These are probabilities for the general adult population and included all four activity levels. In order to adjust the differential risk of

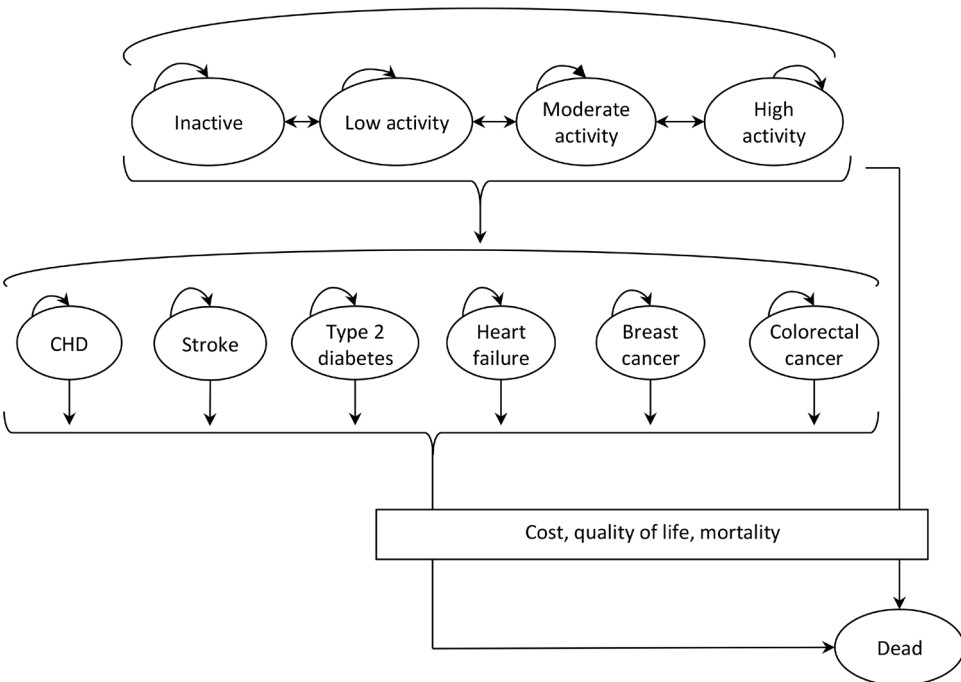

**Figure 1** Conceptual overview of the model. CHD, coronary heart disease.

developing these disease conditions by activity level, we first derived the probability of developing that disease among inactive people, using the method presented by Hurley *et al*.[40] The probabilities for each condition among low, moderate and high activity levels in the cohort were estimated by multiplying the probabilities for the inactive population by corresponding relative risks for low, moderate and high activity.[21–24 26] Online supplementary section A table S1 provides the transitional parameters.

### Mortality rates

All-cause mortality rates by age and gender were derived from the ONS.[29] Mortality consists of disease-specific mortality and mortality due to other causes. We estimated other-cause mortality by subtracting the total number of deaths due to the six disease conditions included in the model from the all-cause mortality total. The other-cause mortality rates by age and gender were assumed constant in the sensitivity analysis. The model assumes that a given proportion of CHD, stroke and heart failure (HF) events would be immediately fatal and people who survived one of these events had an increased subsequent risk of death (online supplementary section A table S2). Case fatality rates for these health states were taken from published population-based studies in the UK.[41–43]

Individuals with type 2 diabetes were assigned an increased risk of mortality using data from a published meta-analysis.[44] Based on standardised mortality ratios reported in long-term follow-up studies of first-ever patients' stroke, a twofold increase in the risk of death after 1 year[45] was applied to the general mortality rates from the life tables to reflect the higher mortality burden postvascular event. Annual mortality rates following the

first HF event were estimated from 10-year case fatality rates in patients admitted with a principal diagnosis of HF in Scotland.[43] The age-adjusted 5-year net survival rates from the ONS[46] were used to estimate an annual risk of cancer death. It was assumed that the mortality rates do not increase due to cancer beyond 5 years after a cancer diagnosis.

### Interventions

We reviewed the literature to identify physical activity interventions targeting the adolescent population. For primarily illustrative purposes, we selected two interventions to test the model and explore the health and economic impact of smaller and greater changes in physical activity. The first was a simple after-school intervention, not costly but likely with smaller benefits,[47] the second a more complex, multicomponent intervention—more costly intervention but with higher expected benefits.[15]

#### After school intervention programme

Mears and Jago[47] included six after-school interventions in their meta-analysis and reported the pooled mean difference of 4.84 (−0.94 to 10.61) min of MVPA per day. These programmes typically included structured or unstructured play, planned MVPA, single or multisport physical activity programme or adhering to specific instructions (eg, maintaining sufficient intensity of exercise during a session).

#### Multicomponent intervention

The intervention effect for school-based multicomponent intervention was taken from a cluster-randomised

**Table 1** Health state utilities and costs used in the model

| Parameter | Value | SE | Distribution | Source |
|---|---|---|---|---|
| Health state utilities | | | | 54 |
| CHD | 0.65 | 0.0203 | Beta (α=357, β=191) | |
| Stroke | 0.52 | 0.0192 | Beta (α=355, β=323) | |
| Heart failure | 0.49 | 0.0194 | Beta (α=326, β=335) | |
| Type 2 diabetes | 0.66 | 0.0054 | Beta (α=5032, β=2548) | |
| Breast cancer | 0.76 | 0.0133 | Beta (α=791, β=256) | |
| Colorectal cancer | 0.67 | 0.0314 | Beta (α=150, β=73) | |
| Costs of health states | | | | |
| CHD first event | £5562 | £556 | Gamma (α=100, β=56) | 48 |
| CHD subsequent | £214 | £21 | Gamma (α=100, β=2) | 48 |
| CHD fatal | £1458 | £146 | Gamma (α=100, β=15) | 48 |
| Stroke first event | £10062 | £1006 | Gamma (α=100, β=101) | 48 |
| Stroke subsequent | £2705 | £270 | Gamma (α=100, β=27) | 48 |
| Stroke fatal | £8805 | £881 | Gamma (α=100, β=88) | 48 |
| HF | £2402 | £240 | Gamma (α=100, β=24) | 49 |
| HF subsequent | £815 | £82 | Gamma (α=100, β=8) | 49 |
| Type 2 diabetes | £1257 | £126 | Gamma (α=100, β=13) | 50 |
| Breast cancer | £12155 | £1215 | Gamma (α=100, β=122) | 51 |
| Colorectal cancer | £16978 | £1698 | Gamma (α=100, β=170) | 52 |

CHD, coronary heart disease; HF, heart failure.

trial[15] implemented in secondary schools in Australia. The intervention included multiple intervention strategies (eg, active physical education lessons, enhanced school sport, supportive school physical activity policies) targeting physical activity. The reported difference in min of MVPA per day between the intervention and control arm at follow-up was 7.0 (2.7–11.4).

### Costs, currency, price date and conversion

The annual costs incurred in each disease health state were based on previously published studies (table 1). First-year costs and subsequent year costs are assigned for each of the health states modelled. Costs of CHD and stroke were taken from the statins health technology assessment.[48] Costs for HF were taken from the UK Prospective Diabetes Study (UKPDS).[49] Costs for type 2 diabetes were based on Diabetes Glycaemic Education and Monitoring trial and included medication and other healthcare costs.[50] Cost for breast cancer was taken from a screening appraisal for breast cancer.[51] The estimated cost is the weighted average treatment costs depending on the prognosis at diagnosis. Cost for colorectal cancer was based on a screening appraisal.[52] The appraisal reported the lifetime cost of colorectal cancer according to the cancer stage, and a weighted average cost was estimated using the proportion of cancers identified at each stage. All the costs were inflated to 2013–2014.[53]

### Health state utility values

Utility weights were used to value a year spent in each of the health states used in the model. A value of 1 means that the health state would be equivalent to full health and 1 year in that state would generate 1 QALY. For example, if an individual spent ten years in the CHD state with a utility of 0.65, they would accrue 6.5 QALYs. The same number of QALYs could be generated by spending 6.5 years in a health state of 1. The lower the utility value, the worse the health state is considered to be. The weights used to value disease health states are given in table 1 and were taken from Sullivan et al,[54] who used UK based community preferences to derive EQ-5D scores from the Medical Expenditure Panel Survey. Utility weights for activity level by age and gender were extracted from the HSE 2012 (table 2).[55]

### Modelling health benefits

We estimated the probability of moving to a higher activity level after intervention by adding intervention-specific MVPA minutes to baseline levels. We then calculated the proportion of cohort members that moved from one activity level to another, and we used this proportion as a transition probability. Members of the cohort who improved physical activity level at the end of cycle 0 were assumed to have a lower probability of developing type 2 diabetes, CHD, stroke, HF or any of the cancers.

**Table 2** Baseline utilities associated with physical activity level

| Age-group | Utility values by activity level | | | | Distribution | Source |
| | Inactive | Low | Moderate | High | | |
| --- | --- | --- | --- | --- | --- | --- |
| 16–34 | 0.897 | 0.918 | 0.937 | 0.943 | Beta | 55 |
| 35–44 | 0.770 | 0.889 | 0.914 | 0.927 | Beta | 55 |
| 45–54 | 0.696 | 0.852 | 0.899 | 0.921 | Beta | 55 |
| 55–64 | 0.648 | 0.861 | 0.863 | 0.907 | Beta | 55 |
| 65–74 | 0.657 | 0.823 | 0.870 | 0.897 | Beta | 55 |
| 65–74 | 0.701 | 0.829 | 0.850 | 0.876 | Beta | 55 |

Inactive: <30; Low: 30–149; Moderate: 150–420 and High: ≥421 MVPA min per week.
MVPA, moderate-to-vigorous physical activity.

### Estimation of the sustainability of intervention effect

The decline in activity levels over time in the 'no intervention'-group was modelled based on data from a recent meta-analysis examining the change in activity level from adolescence to adulthood[8] as well as using prevalence data from the 2012 HSE. The meta-analysis showed a decrease of 6.5 min/day of MVPA in boys and 5.5 min/day of MVPA in girls from adolescence to adulthood. As the review included studies reporting at least one measurement between both 13–19 years and 16–30 years, we assumed that the decrease in MVPA minutes was for a 7-year period. For the intervention group, a 50% decline in intervention effect per year postintervention was assumed. Therefore, the activity levels decreased towards the control activity level after 7 years. The effect of this was that a number of individuals in the intervention group were re-categorised into higher activity groups immediately after the intervention. Over time, many of these individuals would fall back into lower activity groups, and at the end of 7th year, there was no real difference between intervention and control groups. To account for the decline in physical activity occurring with age across the life course, we estimated age-related activity levels using 2012 HSE data on physical activity prevalence by age and gender. Activity levels were estimated in three broad age groups: 24–44, 45–64 and ≥65 years to reflect activity-level differences in adulthood, middle-age and retirement.

### Estimation of costs of intervention

Costs of delivering after-school intervention were taken from a cluster randomised feasibility study in the UK.[56] The authors reported cost estimates (£49 per participant, 2012–2013 price) of a teaching assistant led extra-curricular physical activity intervention. We took that as an indicative cost of the after-school intervention, inflated to 2014 prices (£51 per participant). The cost includes intervention delivery costs, one-off training and non-recurrent costs such as consultation and intervention development work.

Sutherland *et al* performed an economic evaluation alongside the multicomponent intervention[57] and reported that the intervention costed $A394 per participant. This cost includes opportunity costs for delivery of strategies by school staff and community sport and fitness providers,

materials and printing. We converted this cost to 2014-pound sterling (£190 per participant) by applying the gross domestic product deflator index and purchasing power parities conversion rates using the CCEMG-EPPI-Centre Cost Converter (V.1.5).[58] We added the intervention cost in the first year for the intervention groups.

### Validation

The model structure, data sources and the effectiveness evidence used in the exemplar interventions and model results were validated by the study team comprising health economists, behavioural epidemiologists and trialists. Internal validity of the model code was ensured using several tests and by assuming a constant total population throughout the calculations. Furthermore, model predictions were examined to make sure that results from the model were consistent with the model's specifications. We specifically checked lifetime incidence and mortality, as well as physical activity prevalence by age and gender. Details can be found in online supplementary section B figures S1–10.

### Cost-effectiveness analysis

We used NICE reference case[28] and followed existing guidelines for modelling.[59] The analysis was performed from the perspective of the NHS and personal social services. Costs and health outcomes were discounted at 3.5% per year.[28] We estimated the cost-effectiveness ratio for each intervention compared with 'no intervention'. The incremental costs and QALYs gained by the intervention were estimated and averaged across the simulated cohort. The incremental cost-effectiveness ratio (ICER) was estimated as a ratio between the additional expected cost of the intervention, and the additional expected QALYs gained, both relative to the 'no intervention' alternative. The intervention was considered cost-effective if the ICER was no more than the lower NICE recommended threshold of £20 000 per QALY. The uncertainty surrounding the estimates of cost-effectiveness is presented using cost-effectiveness acceptability curves (CEACs). A CEAC shows the probability that an intervention is cost-effective compared with alternative intervention for a range of cost-effectiveness thresholds.[60]

## Sensitivity analyses

We performed sensitivity analyses by changing intervention decay rates and time horizon that affect cost-effectiveness results. Deterministic one-way, scenario and extreme value analyses were undertaken. This was complemented by probabilistic sensitivity analysis (PSA) to assess the combined effects of uncertainty in the input parameters by simultaneously sampling input parameter values from within a specified distribution using Monte Carlo simulations (2000 iterations). Uncertainty about the sustainability of the intervention effects was assessed by varying the decay rates between 0% and 100%. In our base-case analysis, we assumed that the intervention effects are sustained for the first year but decay exponentially at a rate of 50% per annum thereafter, resulting in virtually no intervention effect after 7 years.

Probabilities of disease events and utilities were assumed to follow a beta distribution; costs followed a gamma distribution and risk reductions/HRs a lognormal distribution.[61] The model was developed and implemented in Microsoft Excel.

### Patient and public involvement

Public involvement informed the questions addressed in the overarching research project of which this study is a part. No further public involvement was sought with regards to the development of the research question, the outcome measures or the study design.

## RESULTS

### Base-case analysis

In our base-case analysis, both after-school and multicomponent interventions were associated with higher costs and were more effective than no intervention (table 3). The multicomponent intervention was associated with a QALY gain of 0.002 at an incremental cost of £138, compared with the after-school intervention, yielding an ICER of £68 056.

### Sensitivity analyses

Sensitivity analyses were implemented, varying the base-case assumptions and inputs, as outlined in the methods section (table 3). Both the after-school and multicomponent interventions had more favourable ICERs at lower

**Table 3** Incremental cost-effectiveness ratios in the base-case and sensitivity analyses

| | Total cost (£) | Total QALYs | Incremental cost (£) | Incremental QALY | ICER |
|---|---|---|---|---|---|
| **Base-case analysis** | | | | | |
| No intervention | 4441 | 21.705 | – | – | – |
| After-school intervention | 4491 | 21.710 | 50.64 | 0.004 | 11 486 |
| Multicomponent intervention | 4629 | 21.712 | 137.89 | 0.002 | 68 056 |
| **No (0%) decay of intervention effects** | | | | | |
| No intervention | 4437 | 21.707 | – | – | – |
| After-school intervention | 4451 | 21.898 | 14.32 | 0.191 | 75 |
| Multicomponent intervention | 4571 | 21.987 | 119.59 | 0.089 | 1342 |
| **33% decay of intervention effects** | | | | | |
| No intervention | 4430 | 21.705 | – | – | – |
| After-school intervention | 4479 | 21.718 | 48.86 | 0.013 | 3661 |
| Multicomponent intervention | 4616 | 21.726 | 136.94 | 0.008 | 17 661 |
| **100% decay of intervention effects** | | | | | |
| No intervention | 4432 | 21.705 | – | – | – |
| After-school intervention | 4484 | 21.707 | 51.42 | 0.002 | 28 838 |
| Multicomponent intervention | 4621 | 21.708 | 137.61 | 0.001 | 189 897 |
| **Time horizon (10 years)** | | | | | |
| No intervention | 41 | 7.167 | – | – | – |
| After-school intervention | 92 | 7.170 | 51.47 | 0.004 | 14 204 |
| Multicomponent intervention | 231 | 7.172 | 138.60 | 0.002 | 75 100 |
| **Time horizon (20 years)** | | | | | |
| No intervention | 219 | 12.835 | – | – | – |
| After-school intervention | 270 | 12.838 | 51.09 | 0.004 | 13 414 |
| Multicomponent intervention | 408 | 12.840 | 138.19 | 0.002 | 72 346 |

ICER, incremental cost-effectiveness ratio (incremental cost/incremental QALY); QALY, quality-adjusted life year.

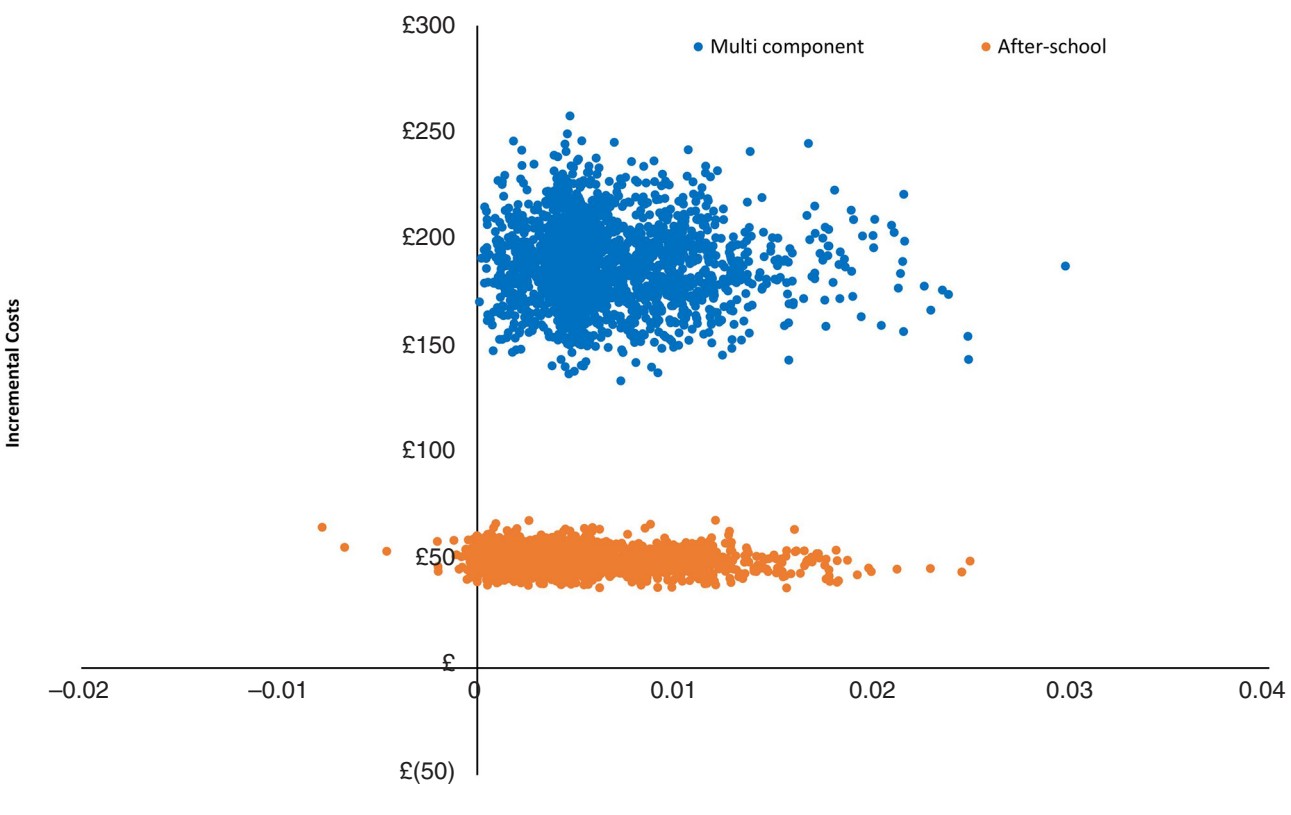

**Figure 2** Scatter plot of incremental costs and QALYs for each intervention, relative to no intervention. QALY, quality-adjusted life-year.

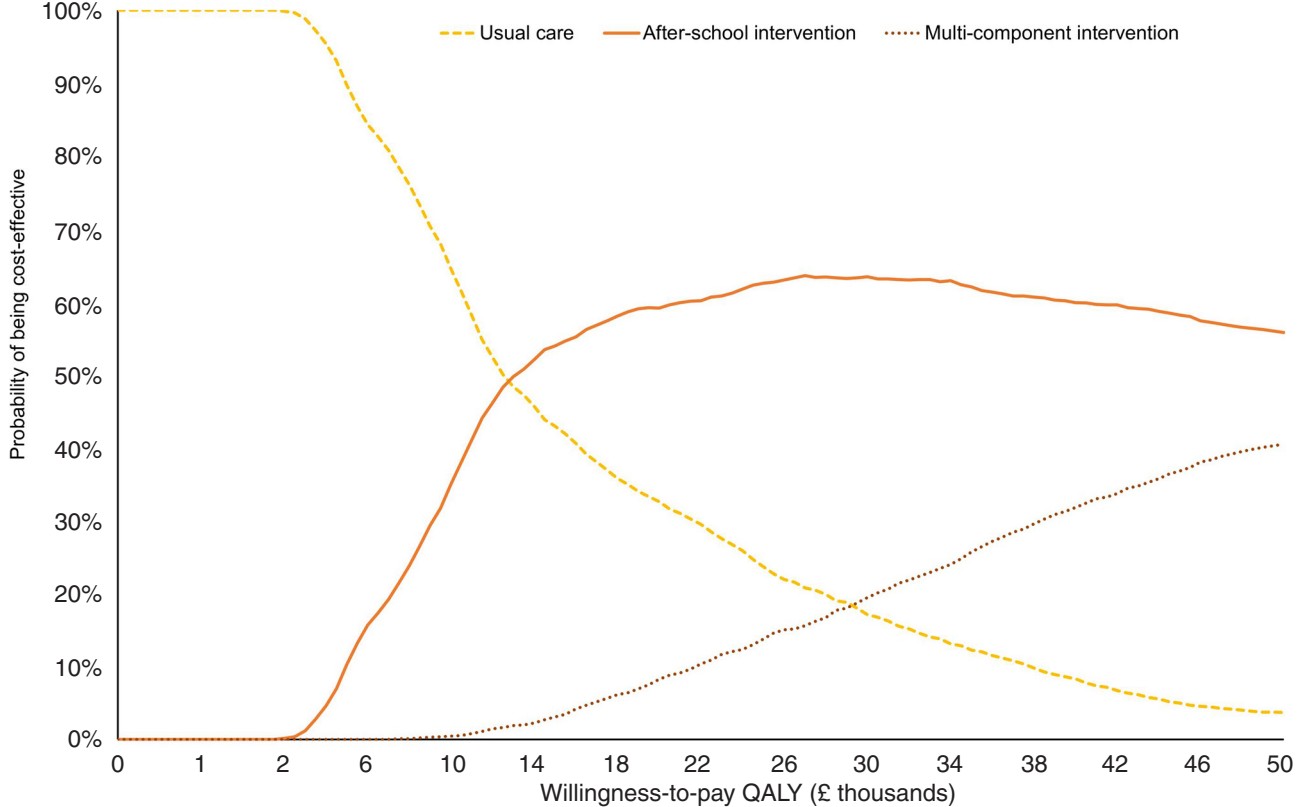

**Figure 3** Cost-effectiveness acceptability curves. QALY, quality-adjusted life-year.

decay rates, indicating that cost-effectiveness of physical activity interventions depends on the sustainability of intervention effects over time. The results of PSA are presented on the cost-effectiveness plane (figure 2), with simulations found to lie predominantly in the northeast quadrant indicating—as expected—an improved health outcome but also at higher spending on physical activity interventions.

Figure 3 depicts the probability of the interventions being cost-effective. At a threshold value of £20 000 per QALY gained, the after-school intervention has the highest probability of being cost-effective (59%) while for the multicomponent intervention this probability is at 8%.

## DISCUSSION
### Main findings
We found that modelling the long-term effects of physical activity among adolescents is feasible, and the model developed here has the potential to estimate the long-term cost-effectiveness of such interventions. The application of the model on two exemplar physical activity interventions in adolescents—one a simple, brief intervention and the other a more complex resource-intensive one—revealed only small differences in terms of lifetime costs per QALYs between the two. Hence, more complex and resource-intensive interventions need not necessarily be better value-for-money in the longer-term compared with cheaper, more targeted approaches. Our findings underline that modelled cost-effectiveness estimates are critically sensitive to assumptions around the sustainability of intervention effects.

### Comparison with previous models
Two previous UK-based modelling studies evaluating the cost-effectiveness of community based physical activity interventions included young people. Although Frew *et al*[20] used the same basic modelling approach, their model included only three activity categories and study participants included both young people and adults (16–70 years old). By contrast, our model has four physical activity categories, included two more health states and health effects are modelled over a lifetime (with a time horizon of 65 years). Pringle *et al*[62] evaluated seven broad categories of community based physical activity interventions (one of which was related to the interventions considered here). Their analysis was based on NICE/Matrix model[63] which used two activity categories (active or inactive) with four disease states. However, they did not focus on young people. Recently, Lee and colleagues[64] modelled the economic and health impact of increasing children's physical activity in the USA. However, unlike in the current model, their model specifically looked at the influence of physical activity on weight status and metabolic profiles and ignored decay in intervention effects or the naturally occurring decline in physical activity associated with ageing.

### Strengths and limitations
Although our model used a similar modelling framework as previous models,[19 20] we included additional health states and focused on adolescent physical activity interventions—this approach has hitherto been neglected, despite the potential importance of intervening at this key stage. We also include the most up to date available evidence on disease conditions. UK-specific incidence rates were used to ensure that patients entering the model match the likely distribution of events in the UK. We chose not to include sedentary behaviour, as there is ongoing debate around its impact on health independent of physical activity.[65]

As with all models, assumptions were required for the analysis. The model presented here is a simplification of a very complex problem. The baseline age of the cohort is 16 years and the effect of physical activity interventions are likely to differ depending on the population age at baseline. We included six disease conditions that have established links with physical (in)activity. This might underestimate the potential impact of physical activity on other disease conditions, most notably mental health. The effect of physical activity on the prevention of depression is still a subject of debate,[66] and a clear dose–response relationship between physical activity and reduced depression is not readily apparent.[67] Further empirical evidence is required to facilitate its inclusion in a future iteration of the model. The current model does not allow for transitions between disease states as this requires more complex modelling. However, this may underestimate the potential impact of physical activity. For example, participants with type 2 diabetes tend to have a higher risk of developing cardiovascular conditions.[68] Although the intervention was aimed at adolescents, due to the nature of the disease conditions included in the model, it would mainly be older adults who develop these diseases.

Our assumption on the decay rates of the additional effect of the intervention, which was based on previous modelling studies,[19 69 70] would mean that there would be very little difference in activity between groups at time points when individuals are starting to develop these diseases. We tested different assumptions on the maintenance of intervention effect to examine influence in cost-effectiveness results. Further research into maintenance of intervention effect would provide valuable information. Our analysis focused on physical activity and only considered direct effects that might result from changes in this health behaviour, while holding any other health behaviours constant. In the real world, physical activity would be expected to interact with other health behaviour choices, in ways that might well affect longer term cost and health outcomes. The existing, very sparse, literature on the interaction between different health behaviours suggest a complex and likely context-specific picture.[64 71]

## CONCLUSION

Interventions to promote physical activity among adolescents represent a potentially promising public health measure to reduce the burden of cardiovascular and other non-communicable diseases. Faced with limited resources, governments need to carefully weigh the costs of any proposed interventions against the associated health benefits expected to be realised over the longer term, in order to ensure that net health gains are maximised. The model developed here has the potential to assess the long-term, beyond trial duration, value-for-money of such interventions. The two purely illustrative applications of the model convey the notion that complex, resource intensive interventions may not necessarily be the ones considered the better buys compared with cheaper, more targeted ones. Maintaining the effect of any behaviour change interventions is challenging as they require personal commitment, encouragement and support over time.

**Twitter** @VijayGC

**Acknowledgements**  We thank Dr Katie Morton and Dr Kirsten Corder for their comments in the early stage of model development work.

**Contributors**  VSG designed the model, performed analysis and wrote the first draft of the manuscript. DT and MS supervised the process, and provided input into data analysis and interpretation of results. AJA and EvS provided critical comments on model structure and data analysis. All authors contributed to the critical revision of the manuscript and approved the final version of the manuscript.

**Funding**  This report is an independent research commissioned and funded by the Department of Health Policy Research Programme (opportunities within the school environment to shift the distribution of activity intensity in adolescents, PR-R5-0213-25001). The views expressed in this publication are those of the author(s) and not necessarily those of the Department of Health. This work was also supported by the Medical Research Council (unit programme number: MC_UU_12015/7). The work was undertaken under the auspices of the Centre for Diet and Activity Research (CEDAR), a UKCRC Public Health Research Centre of Excellence which is funded by the British Heart Foundation, Cancer Research UK, Economic and Social Research Council, Medical Research Council, the National Institute for Health Research, and the Wellcome Trust (MR/K023187/1).

**Competing interests**  None declared.

**Patient consent for publication**  Not required.

**Provenance and peer review**  Not commissioned; externally peer reviewed.

**Data availability statement**  The model was developed using data from publicly available sources, and all the model inputs are described in the paper.

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
