## [Reviewer comments · BMJ Open]

ARTICLE DETAILS

TITLE (PROVISIONAL)	The cost-effectiveness of physical activity interventions in adolescents: model development and illustration using two exemplary interventions
AUTHORS	Gc, Vijay; Suhrcke, Marc; Atkin, Andrew; van Sluijs, Esther; Turner, David

VERSION 1 – REVIEW

REVIEWER	Petri Böckerman University of Jyväskylä Labour Institute for Economic Research IZA
REVIEW RETURNED	06-Dec-2018

GENERAL COMMENTS	Comments 1. The exact contribution to the (international) literature should be stated in the revised introduction.2. What is the external validity of the surveys that are used to provide the estimates regarding the effects of after-school interventions (page 8)?3. Could the incidence figures report the 95% confidence intervals?4. A fundamental issue is that the people face trade-offs regarding the key health behavior choices. Physical activity is time-consuming activity and the time allocated to physical activity is taken from other activities that are potentially correlated with the outcomes of interest in adulthood. This issue should be noted in the revised paper.5. The paper does not consider the indirect effect of physical activity on the economic outcomes in adulthood and the potential joint effects of risky health behaviors including physical inactivity on the economic outcomes (see Böckerman et al. 2018). This issue should be discussed in the revised version of the paper.6. The paper does not consider the potential heterogeneity in the estimated effects. The relationships can differ significantly e.g. by gender.7. The concluding section of the paper should discuss more about the practical policy lessons that can be drawn from the estimation results. Reference Böckerman, P., Hyytinen, A., Kaprio, J., & Maczulskij, T. (2018). If you drink, don't smoke: Joint associations between risky health behaviors and labor market outcomes. Social Science and Medicine, 207, 55-63.
---

REVIEWER	Atif Adam Johns Hopkins Bloomberg School of Public Health , USA
REVIEW RETURNED	04-Feb-2019

GENERAL COMMENTS	1. The authors use the term “adolescent” were loosely in the paper. While the ages of adolescence range from 10 to 19 years, the simulation focuses on 16 years on. This should be made clear. 2. The intervention PA models are well tested in the simulation, but the paper seems to talking more about the model and less about the results. The title and premise go to imply that the paper will evaluate and convey an evaluation of PA among adolescents. This is misleading as the results are poorly explained and the discussion is sparse. 3. The Markov model in the paper purely evaluates the risk reduction from PA on health conditions over the lifetime. While the risk estimates are all referenced and established, it still paints a simplistic view on changing health states over the life time and the associated costs. There is clear evidence that cardio-metabolic factors (hypertension, hyper-lipedema) along with obesity status play a major in role in risk of obesity-associated conditions (CVD, T2DM, Stroke etc etc). Additionally, there have been economic evaluations showing that incorporating the life-time costs from cardio-metabolic factors and obesity changes are key in understanding savings from interventions. Purely focusing on the end-term health costs like stroke, CVD etc, while appropriately have higher yearly medical costs at the individual level, drastically underestimates the cost savings at the population level from reduction in other costs over the life time. 4. Please expand limitation section on clinical model implication, implication on productivity savings etc. 5. please revise the paper to focus on a more through evaluation on PA interventions in adolescents (16 -19 years) with specific areas of ICER and perspectives.
--

VERSION 1 – AUTHOR RESPONSE

Response to Reviewers’ Comments:

Reviewer: 1

1. The exact contribution to the (international) literature should be stated in the revised introduction.

We have now revised the last two paragraphs of the introduction section and state the potential contribution of our manuscript to the existing literature. The revised text reads as:

“Furthermore, much of the health benefits of physical activity interventions occurs in the future. Also, many interventions are focused on adult or elderly populations. The long-term costs and health benefits of physical activity interventions in adolescent population are a comparatively scarcely researched area. To fill this critical research gap, we developed a decision analytic model aimed at quantifying the potential long-term costs and quality-adjusted life-year (QALY) implications of changes in activity levels during adolescence. We then illustrate some of the practical implications of taking a longer-term perspective by applying the model to two exemplary intervention programmes to show how the changes in levels of adolescent physical activity could affect activity levels throughout lifetime, as well as the resulting longer term costs and health benefits.”

Please see also our related response to point 2 of reviewer 2 below.

2. What is the external validity of the surveys that are used to provide the estimates regarding the effects of after-school interventions (page 8)?

We used effectiveness data for after-school interventions from the Mears & Jago (2016) meta-analysis. Their meta-analysis included six studies, of which five used an objective measure of MVPA (accelerometry). Moreover, four of the six studies included in the meta-analysis were from the US. Although the estimate from the meta-analysis is robust, we agree that due to methodological differences between studies (e.g. in terms of physical activity measurement and context characteristics), there are limits to the generalisability of the cost-effectiveness results across different settings. Please note though that our use of one particular type of school-based adolescent PA intervention (i.e. after-school interventions) is primarily for illustrative purposes to show how our model could be applied to one specific (out of many potential) interventions.

Mears R, Jago R Effectiveness of after-school interventions at increasing moderate-to-vigorous physical activity levels in 5- to 18-year olds: a systematic review and meta-analysis. *Br J Sports Med* 2016; 50: 1315-1324.

3. Could the incidence figures report the 95% confidence intervals?

We have provided 95% confidence intervals of relative risk (RR) estimates by physical activity category (Table S1). These RR estimates were used to adjust disease incidence using the method proposed by Hurley & Matthews (2007) which was cited in the main text (under Transition probabilities sub-heading, page 7).

Hurley SF, Matthews JP The Quit Benefits Model: a Markov model for assessing the health benefits and health care cost savings of quitting smoking. *Cost Eff Resour Alloc* 2007; 5: 2.

4. A fundamental issue is that the people face trade-offs regarding the key health behavior choices. Physical activity is time-consuming activity and the time allocated to physical activity is taken from other activities that are potentially correlated with the outcomes of interest in adulthood. This issue should be noted in the revised paper.

We considered two exemplar physical activity interventions to explore potential long-term costs and health benefits of targeting one health behaviour only, i.e. physical activity. We agree that in an ideal setting, we would be able to take into account in our model how physical activity does or does not interact with other relevant health behaviours to produce longer term outcomes. However, to date the evidence is very limited as to the exact nature of the interaction among different health behaviours. We have revised the discussion section along with the response to comment 5 below.

5. The paper does not consider the indirect effect of physical activity on the economic outcomes in adulthood and the potential joint effects of risky health behaviors including physical inactivity on the economic outcomes (see Böckerman et al. 2018). This issue should be discussed in the revised version of the paper.

As mentioned above in response 4, our analysis focused on one health behaviour (physical activity), and the reviewer is correct that the analysis did not consider the indirect effects. Furthermore, we are unaware of cost-effectiveness modelling studies that have tried to take interaction between physical activity and other health behaviour choices explicitly into account. Hence, we saw it as beyond the scope of present paper to do so in our merely illustrative exercise. We have added the following sentences under study limitation on page 18:

“Our analysis focused on physical activity and only considered direct effects that might result from changes of this health behaviour, while holding any other health behaviours constant. In the real world, physical activity would be expected to interact with other health behaviour choices, in ways that might well affect longer term cost and health outcomes. The existing, very sparse, literature on the interaction between different health behaviours suggests a complex and likely context-specific picture.[64,71]”

[64] Lee BY, Adam A, Zenkov E et al. Modeling The Economic And Health Impact Of Increasing Children's Physical Activity In The United States. *Health Aff (Millwood)* 2017; 36: 902-908.

[71] Böckerman P, Hyytinen A, Kaprio J et al. If you drink, don't smoke: Joint associations between risky health behaviors and labor market outcomes. *Soc Sci Med* 2018; 207: 55-63.

6. The paper does not consider the potential heterogeneity in the estimated effects. The relationships can differ significantly e.g. by gender
Based on the previous meta-analysis, we incorporated the differential change in activity level from adolescent to adulthood, i.e. 6.5 and 5.5 minutes of MVPA per day in boys and girls respectively. However, we did not consider the differential effectiveness of physical activity intervention by gender. A recent meta-analysis of school-based cluster RCTs with accelerometer-assessed activity (Love et al. 2019) showed no evidence of differential effectiveness by gender or socio-economic position. There could potentially be a differential effect of physical activity on health outcomes, but due to the lack of clear evidence pointing in this direction, we felt our decision to abstract from such differences for the purpose of the paper would be legitimate.
Love R, Adams J, van Sluijs EMF Are school-based physical activity interventions effective and equitable? A meta-analysis of cluster randomized controlled trials with accelerometer-assessed activity. *Obes Rev* 2019. DOI: 10.1111/obr.12823

7. The concluding section of the paper should discuss more about the practical policy lessons that can be drawn from the estimation results.
The primary purpose of this paper is to develop a decision analytic model, which we then use and apply to two (somewhat arbitrarily selected) interventions, in order to illustrate some of the practical implications of taking a longer-term perspective while quantifying potential long-term economic and health benefits of the increases in activity level during adolescence. The specific nature of the interventions we selected is not the primary focus of our paper, and hence we would prefer not to assign too big policy implications to those specific interventions. We have now revised the introduction section to make this more clear early on, and we have amended the title of the paper (see also our response to comment 2 of Reviewer 2 below).

Reviewer: 2

1. The authors use the term “adolescent” were loosely in the paper. While the ages of adolescence range from 10 to 19 years, the simulation focuses on 16 years on. This should be made clear.
We agree that the period between 10 to 19 years of age is generally considered adolescent. Recently there has been a suggestion to increase this range up to 24 years. It is argued that 10-24 years corresponds more closely to adolescent growth and popular understanding of this life phase (Sawyer et al. 2018). In our analysis, we used 16 years as a mid-point value. We have added a sentence in the Discussion section to clarify this:
“The baseline age of the cohort is 16 years and the effect of physical activity interventions are likely to differ depending on the population age at baseline.”
Sawyer SM, Azzopardi PS, Wickremarathne D et al. The age of adolescence. *Lancet Child Adolesc Health* 2018; 2: 223-228.

2. The intervention PA models are well tested in the simulation, but the paper seems to talking more about the model and less about the results. The title and premise go to imply that the paper will evaluate and convey an evaluation of PA among adolescents. This is misleading as the results are poorly explained and the discussion is sparse.
The focus of our paper was to describe the development of a PA model to quantify the potential long-term costs and health benefits of PA interventions among adolescents. The PA interventions presented are intended as purely illustrative examples of how the model could be applied. To reflect

this, we have modified the title of the paper which now reads “The cost-effectiveness of physical activity interventions in adolescents: model development and illustration using two exemplary interventions”. We have also made the purpose of the paper more clear in the introduction now (see our response to comment 7 of reviewer 1 above).

3. The Markov model in the paper purely evaluates the risk reduction from PA on health conditions over the lifetime. While the risk estimates are all referenced and established, it still paints a simplistic view on changing health states over the life time and the associated costs. There is clear evidence that cardio-metabolic factors (hypertension, hyper-lipedema) along with obesity status play a major in role in risk of obesity-associated conditions (CVD. T2DM, Stroke etc etc). Additionally, there have been economic evaluations showing that incorporating the life-time costs from cardio-metabolic factors and obesity changes are key in understanding savings from interventions. Purely focusing on the end-term health costs like stroke, CVD etc, while appropriately have higher yearly medical costs at the individual level, drastically underestimates the cost savings at the population level from reduction in other costs over the life time.

We agree that the model purely evaluated the risk reduction from PA on health conditions over the lifetime and the effect of increase PA is not mediated through the reduction in other risk factor values such as hypertension. It is the case that this approach may underestimate the true benefits of PA interventions in terms of both health and economic impact. The model presented here is a simplification of a very complex situation and relationship. We have added a sentence in the discussion section along with the response to reviewer 1, and we have now cited Lee et al. 2017. Lee BY, Adam A, Zenkov E et al. Modeling The Economic And Health Impact Of Increasing Children's Physical Activity In The United States. Health Aff (Millwood) 2017; 36: 902-908.

4. Please expand limitation section on clinical model implication, implication on productivity savings etc.

As the reviewer suggests, we have expanded the limitation section to incorporate issues around the effect of PA on other risk factors, indirect effects etc.

5. please revise the paper to focus on a more through evaluation on PA interventions in adolescents (16 -19 years) with specific areas of ICER and perspectives.

As described above in response #2 (and response #7 of reviewer 1), the aim of the paper was to describe the development of a model to quantify the potential long-term health and economic implications of changes in PA levels during adolescence. We used exemplary PA interventions to explore the model and highlight the practical implications of taking a long-term perspective. We have rephrased the title and aim of the paper to make this more obvious.

VERSION 2 – REVIEW

REVIEWER	Petri Böckerman University of Jyväskylä Finland
REVIEW RETURNED	26-Apr-2019

GENERAL COMMENTS	I am happy with the revised version of the paper. I like the research question, the structure of the paper, the quality of writing, and the way the authors describe their empirical proceeding and results. Most importantly, the authors have addressed all the issues stated in my referee report for the first version appropriately.
---

REVIEWER	Atif Adam JHU, USA
REVIEW RETURNED	13-May-2019

GENERAL COMMENTS	The authors have addressed the comments previously noted. Things to note in this review are primarily to reflect the overall purpose of the paper. The authors have noted “ The primary purpose of this paper is to develop a decision analytic model, which we then use and apply to two (somewhat arbitrarily selected) interventions, in order to illustrate some of the practical implications of taking a longer-term perspective while quantifying potential long-term economic and health benefits of the increases in activity level during adolescence.” “The specific nature of the interventions we selected is not the primary focus of our paper, and hence we would prefer not to assign too big policy implications to those specific interventions.” Keeping this in mind the Main Findings (Page 16 Line 33-43) still reads as the main outcomes of the paper are modelling the long-term effects of PA and CEA estimates. As the authors have pointed out, that is not the focus of this paper. Please revise.
--

VERSION 2 – AUTHOR RESPONSE

Response to Reviewers' Comments:

Reviewer: 1

I am happy with the revised version of the paper. I like the research question, the structure of the paper, the quality of writing, and the way the authors describe their empirical proceeding and results. Most importantly, the authors have addressed all the issues stated in my referee report for the first version appropriately.

We are happy that the reviewer finds the changes acceptable. Thank you for the review, helpful comments and kind remarks on our paper.

Reviewer: 2

The authors have addressed the comments previously noted.

Things to note in this review are primarily to reflect the overall purpose of the paper. The authors have noted “ The primary purpose of this paper is to develop a decision analytic model, which we then use and apply to two (somewhat arbitrarily selected) interventions, in order to illustrate some of the practical implications of taking a longer-term perspective while quantifying potential long-term economic and health benefits of the increases in activity level during adolescence.”

“The specific nature of the interventions we selected is not the primary focus of our paper, and hence we would prefer not to assign too big policy implications to those specific interventions.”

Keeping this in mind the Main Findings (Page 16 Line 33-43) still reads as the main outcomes of the paper are modelling the long-term effects of PA and CEA estimates. As the authors have pointed out, that is not the focus of this paper. Please revise.

Thanks indeed for the positive, constructive response. We have made revisions on the main findings section (page 16) according to your comments and suggestions. The revised paragraph now reads:

“We found that modelling the long-term effects of physical activity among adolescents is feasible, and the model developed here has the potential to estimate the long-term cost-effectiveness of such interventions. The application of the model on two exemplar physical activity interventions in adolescents – one a simple, brief intervention and the other a more complex resource-intensive one – revealed only small differences in terms of lifetime costs per QALYs between the two. Hence, more complex and resource-intensive interventions need not necessarily be better value-for-money in the longer-term compared to cheaper, more targeted approaches. Our findings underline that modelled cost-effectiveness estimates are critically sensitive to assumptions around the sustainability of intervention effects.”

In addition to the comments from reviewers, we have corrected a typo by replacing ‘chronic heart disease’ with ‘coronary heart disease’ (page 4, line 35) and added ‘type 2 diabetes’ (page 11 line 40) to include the fact that improved physical activity levels at the end of cycle 0 were assumed to have a lower probability of developing six disease conditions modelled, i.e. type 2 diabetes, CHD, stroke, HF, breast and colorectal cancers.